# Numerical Analysis of Ultrasonic Nebulizer for Onset Amplitude of Vibration with Atomization Experimental Results

**Yu-Lin Song [1,2,*], Chih-Hsiao Cheng [3] and Manoj Kumar Reddy [2]**

[1] Department of Bioinformatics and Medical Engineering, Asia University, Taichung 413, Taiwan
[2] Department of Computer Science and Information Engineering, Asia University, Taichung 413, Taiwan; manoj14a0kumar@gmail.com
[3] Zeng Hsing Industrial Co., Taichung 411, Taiwan; forcea9931@gmail.com
[*] Correspondence: d87222007@ntu.edu.tw

**Abstract:** In this study, the onset amplitude of the initial capillary surface wave for ultrasonic atomization of fluids has been implemented. The design and characterization of 485 kHz microfabricated silicon-based ultrasonic nozzles are presented for the concept of economic energy development. Each nozzle is composed of a silicon resonator and a piezoelectric drive section consisting of three Fourier horns. The required minimum energy to atomize liquid droplets is verified by COMSOL Multiphysics simulation software to clarify experimental data. The simulation study reports a minimum vibrational amplitude (onset) of 0.365 μm at the device bottom under the designated frequency of 485 kHz. The experimental study agrees well with the suggested frequency and the amplitude concerning the corresponding surface vibrational velocity in simulation. While operating, the deionized water was initially atomized into microdroplets at the given electrode voltage of 5.96 V. Microdroplets are steadily and continuously formed after the liquid feeding rate is optimized. This newly designed ultrasonic atomizer facilitates the development of capillary surface wave resonance at a designated frequency. A required vibrational amplitude and finite electric driving voltage promote not only the modern development in the green energy industry, but also the exploration of noninvasive, microencapsulated drug delivery and local spray needs.

**Keywords:** capillary waves; surface wave; subharmonic; resonance; COMSOL; ultrasonic atomizer

## 1. Introduction

In 1962, Dr. Robert Lang reported that atomization of fluids was caused by a capillary wave formed in an unstable state and small droplets were formed by a collapsing of unstable surface waves. Dr. Robert Lang's work proves the correlation between his atomized droplet sizes relative to Rayleigh's liquid wavelength [1]. Ultrasonic atomized nozzles use high-frequency vibration produced by piezoelectric crystal driving transducers acting upon the nozzle tip that will create capillary waves in a liquid film. Once the amplitude of the capillary surface waves reaches a critical point (due to the power supplied by the generator), surface waves become too tall to support each other and tiny droplets fall off the tip of each wave, resulting in atomization. The basic performances of such devices have been studied experimentally by Lang (1962), Griesshammer and Lierke (1967), Stamm and Pohlman (1965), Sindayihebura et al. (1997), and Topp (1973). The factors that influence the initial droplet size produced are surface tension, frequency of vibration, and viscosity of the liquid. Beyond the range of human hearing, the smallest drop size produces the highest frequencies. Ultrasonic nozzles were first commercialized by Dr. Harvey L. Berger in "Fuel burner with improved ultrasonic atomizer", published 21 January 1975, USA 3861852.

The ultrasonic atomizer is a useful technology and has many practical applications. The ultrasonic atomizer has been successfully applied in areas of biomedical engineering, such as in food industries, chemical coatings, and miniature carriers in pharmaceuticals, as well as for the fabrication of printed electronics and sensors [2,3]. Propagations of ultrasonic surface waves were of great interest in the middle of the 20th century, as they have enabled atomization of incompressible frictionless fluid and have often been known in compact, atomized liquid droplets [4,5]. The control of the atomized droplet process and the characterization of the vaporized droplet size are important issues in obtaining good physical properties and repeatability [6,7]. Moreover, biomedical engineering has successfully applied the technique in drug delivery, pharmacy, and gene therapy [8,9]. Several acoustic microscopy applications were extended to the nondestructive testing of a small structure, intravascular imaging, and ultrasound contrast agent imaging [10,11].

This work develops a practical device for ultrasonic atomization; we explain an onset vibrational amplitude under a designated frequency at the device's bottom [12]. We also aim to know more about the initial surface condition when the energy of ultrasonic atomization of fluids meets the minimum requirement to make a potential green device [13]. The numerical analysis and atomization experimental progress confirmed the theoretical concept to yield tiny atomized droplets under a certain vibrational stimulus [14]. However, the theoretical background was conducted in a steady state. The formation of atomized droplets should be a steadfast and continuous ultrasonic atomization progress. It is necessary to understand more about the dynamic behavior in ultrasonic atomization for real applications [15]. In this study, a dedicated software COMSOL Multiphysics 5.4 is utilized to understand the formation of atomized droplets by the computational fluid dynamics method. The velocity field distribution is solved by finite-element analysis. With regard to the Faraday instability wave, if the effect of the lateral boundary is ignored, the fluctuations generate a standing wave at the surface because of vibration at the bottom. This supports the critical vibrational amplitude (i.e., the characteristic onset of vibrational amplitude) and indicates the constitutive atomized droplets. Further, as the prediction results, we represent the fabrication of the Fourier horn nozzle in three sets at a 485 kHz designated frequency that produces small atomized droplets around the nozzle tip. The feeding liquid flow rate and its vibrational amplitude were carefully measured and statically calculated to understand the manufacturing reliability.

## 2. Governing Equation

Examinations of surface wave amplification began from the view of Faraday instability in 1831 [16]. Exactly when a vertical vibration is applied for some liquid water on a planar substrate, surface waves are surrounded on a stationary water surface. The destabilized even water surface is accumulated by gatherings of nonlinear standing waves. It is noticed that the periodicity of the working vibration of the capillary surface wave is doubled. In 1871, Kelvin proposed the outstanding condition of capillary surface wave amplification [17]:

$$\lambda = [2\pi T / (\rho f_s^2)]^{1/3} \tag{1}$$

where $\lambda$ is the wavelength of the capillary waves, T is the surface tension, $\rho$ is the liquid density, and $f_s$ is the frequency of the capillary surface waves.

Rayleigh changed Equation (1) in 1883 as follows:

$$\lambda = [8\pi T / (\rho f^2)]^{1/3} \tag{2}$$

where $f$ is the forcing sound frequency that is observationally two times $f_s$ from practical results.

Benjamin used Mathieu's condition to build a stability diagram for an ideally incompressible frictionless liquid [18]. If a characteristic number and an amplitude of vibration (p, q) lie in any concealed area of the stability diagram as shown in Figure 1, Mathieu's

equation would lead to the unstable region that gives rise to the conspicuous standing wave on the fluid surface. If (p, q) situates in one of the unshaded areas, the solution is in a stable district, which implies that the liquid fluid surface is thoroughly quiet without consistent periodic waves.

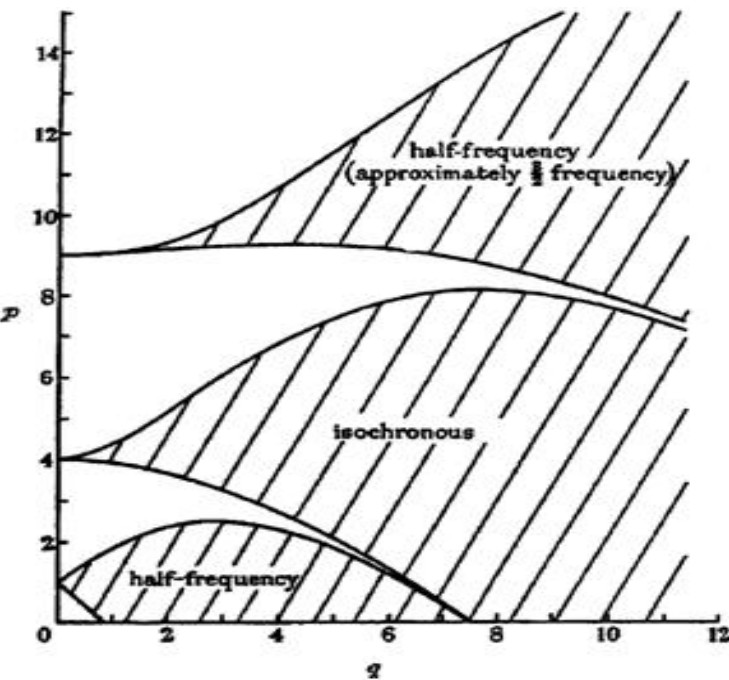

**Figure 1.** Stability chart for the solutions of Mathieu's equation. Reprinted from Benjamin and Ursell [18].

Eisenmenger then built up a technique for estimating a surface viscosity by methods for rheolinear oscillations [19]. Mathieu equation was conducted by the variable transformation. The minimum capillary wave amplitude is

$$h_{cr} = 2\nu \left[\rho /(\text{T}\omega_s)\right]^{1/3} \tag{3}$$

where $\omega_s = (\text{T} k^3/\rho + gk)^{1/2}$, $\nu$ is the kinematic viscosity of the fluid, $k$ is the capillary wave number, and g is the gravity.

Under low surface viscosity of watery arrangements and feeble vessel boundary conditions, the deliberate capillary surface wave amplitude is half the excitation amplitude with $\pi/4$ phase delay [20]. The experimental approach showed a few frequency vibrations in the range from 2 kHz to 800 kHz. The size of the atomized droplet and the capillary wavelength at a given frequency is proposed as

$$\text{D} = 0.34\lambda \tag{4}$$

where D is the diameter of the atomized droplet.

To investigate the formation of atomized droplets, Prof. Perron observed fluid atomization and found that uniform water droplets are formed under a sufficient oscillatory vibration [21]. Prof. Sindayihebura utilized a high-velocity camera technique at high magnification to identify the atomization behavior at the working frequencies of 32 and 50 kHz [22]. The ultrasonic atomized droplet size and velocity distribution of a methanol and glycerol solution were theoretically studied. The ratio between the mean atomized droplet diameter and the vibration wavelength is usually at 0.34–0.36. Prof. Kumar noted that the preferred wave number in the Faraday instability is primarily determined through a Rayleigh–Taylor instability in the deep-water limit (wavelength << liquid depth) by performing linear stability calculations [23].

Ultrasonic atomization studies were for the most part centered on the use of low frequency. For the ultrasonic atomization at a higher frequency, the base required working amplitude remains a provoking topic for researchers investigating surfactant-covered liquid and developing energy-saving devices [24,25].

## 3. Finite-Element Analysis

COMSOL Multiphysics 5.4 is an advanced software of limited component investigation. It was utilized to demonstrate the basic vibrational amplitude of the silicon-based three-horn Fourier nozzle and the velocity of the surface wave before tests in the following area. A turning hardware model and a laminar stream highlight are referenced for numerical reproductions of surface wave amplification. Navier–Stokes condition is the default mathematic resolver, and the nonlinear impact of liquid fluid is considered to direct arrangements. The vibrational frequency of the silicon-based three-horn ultrasonic nozzle was characterized at f = 485 kHz in the base, at which the characteristic onset amplitude of the capillary surface wave (i.e., the required minimum energy) is calculated for an ultrasonic atomizer. The acquired data support the geometry structure of nozzle detail. The geometry in this reproduction has been characterized in a past investigation [26]. Our endeavor here is to recognize the base plentifulness of atomization under a longitudinal sinusoidal vibration from a nozzle bottom. The investigation case in this report is fluid deionized water, which is broadly useful in numerous applications.

To clarify the onset amplitude of surface waves under a longitudinal sinusoidal oscillation in a liquid-filled nozzle, a moving wall is the setting for generating waves at the nozzle bottom. The velocity in the longitudinal direction is $h_e \times \omega \times \sin(\omega t)$.

The momentum equation is [27]:

$$\rho \frac{\partial u}{\partial t} + \rho(u \cdot \nabla)u = \nabla \cdot \left[ -PI + \mu(\nabla u + (\nabla u)^T) - \frac{2}{3}\mu(\nabla \cdot u)I \right] - \rho(g - h_e \omega^2 \sin \omega t) \tag{5}$$

where $u$ is the velocity field, $\rho$ = 1000 kg/m³ is the homogeneous density, $\omega = 2\pi f$ is the angular frequency, $\mu$ = the dynamic viscosity, $h_e$ = the given amplitude, $P$ = the equilibrium stress at the free fluid surface, and $I$ = the 3 × 3 identity matrix.

There are at least six components inside a vibrating wavelength. Free surface limit is the default with the goal that the velocity of the liquid fluid surface can be removed from a limited distinction estimate technique at the middle region. The free surface at the fluid, fluid surface, was determined as follows [28]:

$$\left[ -PI + \mu(\nabla u + (\nabla u)^T) - \frac{2}{3}\mu(\nabla \cdot u)I \right] \cdot n = -P_a + T(\nabla_t \cdot n)n - \nabla_t T \tag{6}$$

where $P_a$ is the atmospheric pressure of 1 atm. The full theoretical conduction can be found in the supplementary material.

Because $\frac{\partial \rho}{\partial t} + \nabla \cdot (\rho u) = 0$ and $\nabla \cdot u = 0$ for the zero consumption of mass, Equation (5) can be rewritten as

$$\rho \frac{\partial u}{\partial t} + \rho(u \cdot \nabla)u = -P_a + T(\nabla_t \cdot n)n - \nabla_t T - \rho(g - h_e \omega^2 \sin \omega t) \tag{7}$$

The velocity is expanded with an increased number of periods. In the inset in Figure 2, the schematic outline shows the vibrational amplitude of 0.365 μm at the nozzle tip as demonstrated. In the interim, the velocity amplifies on the droplet surface as set apart by red shading [29].

To estimate the onset vibrational amplitude, a scope of vibrational frequency $f$ of the silicon-based three-horn ultrasonic nozzle was filtered by COMSOL. Figure 2 represents how the capillary surface wave is upgraded inside 400 periods under an onset vibrational amplitude at 0.365 μm. The minor preformed droplets leave the temperamental surface

[30] at the most extreme velocity of 0.06 m/s. The vibrational amplitude is upgraded at the center of the Fourier horns by the given vibrational amplitude value.

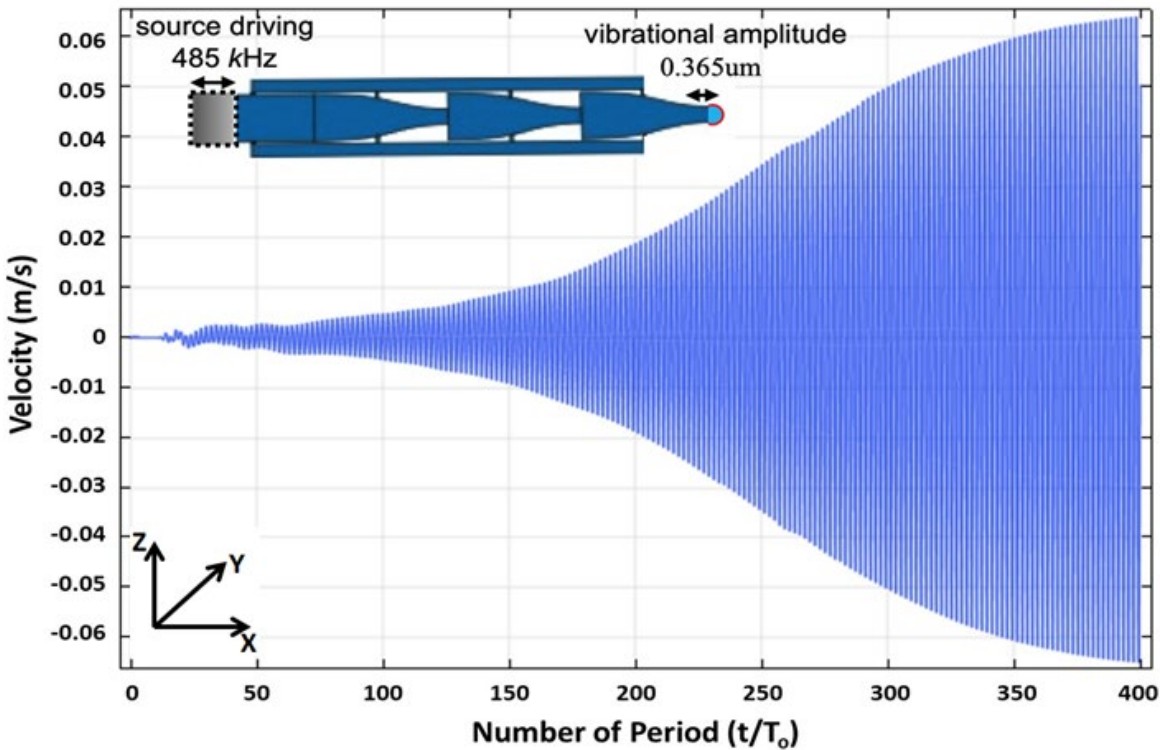

**Figure 2.** The modeling plot of the velocity in the Z-axis versus time at the nozzle tip of the silicon-based three-horn.

In this case, based on the water, we use water parameters of density $\rho$ = 1000 (kg/m³), kinematic viscosity coefficient $\nu$ = 1.02 × 10⁻⁶ (m²/s), and surface tension T = 7.2 × 10⁻² (N/m). The frequency range was plotted for the perception of harmonic frequency and the subharmonic frequency at 485 and 242.5 kHz individually (Figure 3). It shows that the beginning vibrational amplitude occurred at the assigned frequency of 485 kHz. Further, the wavelength of the capillary surface wave is 19.4 μm, as determined from Figure 4 by the peak-to-peak separation. It is fascinating that the simulated value of 19.4 μm and the theoretical value of 19.8 μm (i.e., given Equation (2)) for the wavelength of the capillary wave are comparable [31].

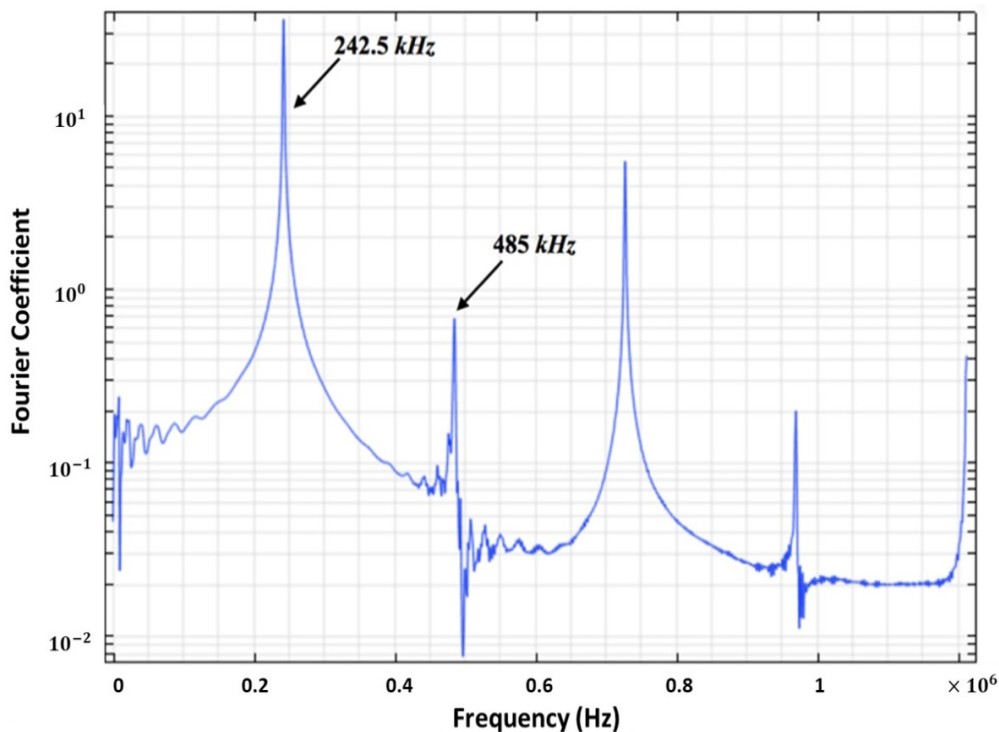

**Figure 3.** The calculated frequency spectrum at the tip of the silicon-based three-horn. The designated harmonic frequency from COMSOL Multiphysics 5.4 is 485 kHz. As expected, 242.5 kHz is the corresponding subharmonic frequency.

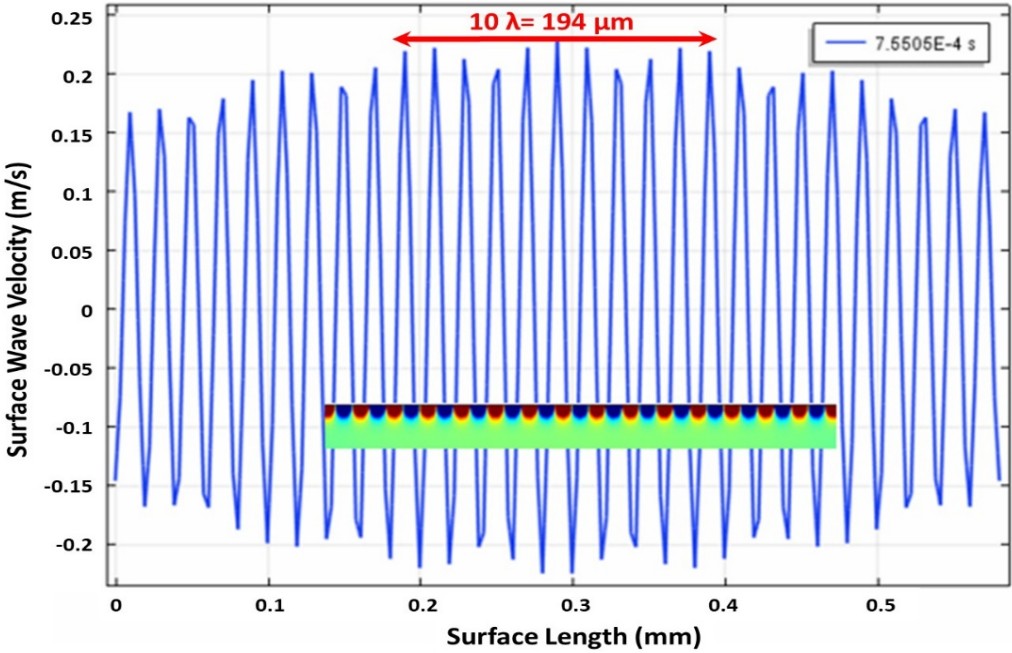

**Figure 4.** The vibrational velocity is propagated throughout the water surface on the tip of the silicon-based three-horn in the modeling. Inset: the 2D velocity profile at the corresponding position at $7.5505 \times 10^{-4}$ s simulation time. The red and blue colors present the displacement in the opposite directions.

The lateral velocity profile along Z-axis can be found in the inset of Figure 4 covering the positional velocity of the capillary surface wave. In the supporting material for vibra-

tional velocity (S1) and vibrational amplitude (S2), it can be seen that the vibrational amplitude working at 0.40 μm has a critical upgrade of capillary surface wave propagation. In the supporting material S3, the uprooting and discharge of supporting fluid deionized water along the X-axis are clearly seen. The waves began from PZT plates in the base areas and were collected. Until the beginning vibrational amplitude of 0.365 μm was met, the necessary surface pressure was discharged to atomize modest fluid droplets. Interestingly, beneath the beginning amplitude of 0.365 μm, the capillary surface wave cannot be energized at 0.36, μm as can be found in the supporting material for vibrational velocity (S4) and vibrational amplitude (S5).

## 4. Experiments

The resonant frequency and its onset amplitude of the capillary surface wave were validated here to help the modeling results. Exploiting the microelectromechanical system (i.e., MEMS), the silicon-based three-horn ultrasonic nozzle was created as delineated in Figure 5a with a focal channel for liquid flow [32]. Three separate Fourier amplifiers were connected in series into one. The magnification obtained by this amplifier can be described in three sections to obtain 8 times magnification. The displacement of the three sections at 500 kHz is in longitudinal vibrational mode. The detailed system and preparation method for the MEMS-based Fourier horn nozzle are described below, including the base areas where PZT plates are to be bounded. The lengths of the silicon-based three-horn and the PZT plates in the transducer drive area were changed by acquiring an ideal high full frequency [33]. The transducer driving square and the resonator square were stuck together to frame a nozzle with a focal channel for liquid trespassing. As deionized water enters the focal channel of the nozzle in 200 μm × 200 μm, a curved thin film remained at the nozzle tip that vibrated at the reverberation frequency of 485 kHz, bringing about the development of standing capillary waves. At the point when the accumulation of the standing capillary waves exceeds a threshold of fluidic surface strain, a shower of droplets/mist is normally created. The normal droplet size from the atomizer was assessed as 7 μm by laser diffraction procedure [34].

Every silicon-based resonator is associated with an arrangement with half-wavelength of ultrasound. The cross-sectional region of the Fourier horn changes bit by bit from the channel to the outlet zone. At the point when the PZT transducer drives a sinusoidal swaying through the Fourier horns, a standing wave is created by the longitudinal vibration of the nozzle tip. Because of energy protection, the vibrational amplitude of the longitudinal direction is collected [35]. Longitudinal vibration was dictated by the Doppler frequency shift in a test estimation of the vibrational frequency and sufficiency at the nozzle tip. Considering working voltages that are changed to mechanical energies, the vibrational amplitude of the three-horn nozzle increased by $2^3$ times. As the greatest longitudinal vibration breaks the surface pressure of the liquid, a temperamental fluid capillary surface wave, in the long run, transforms into modest atomized droplets. The progress of the atomized droplets can be seen in Figure 5b–e.

Deionized water was selected as the applied fluid for its adaptable dissolution of different synthetic chemicals. After the underlying atomization of deionized water, the ultrasonic atomized droplets were consistently shaped under a specific sustaining rate, as can be found in the supporting material S6. Capillary surface waves were separated at the droplet of deionized water. The three-horn ultrasonic nozzle was operated at 484.5 kHz with the applied voltage Vp between 6.5 and 7.9 V.

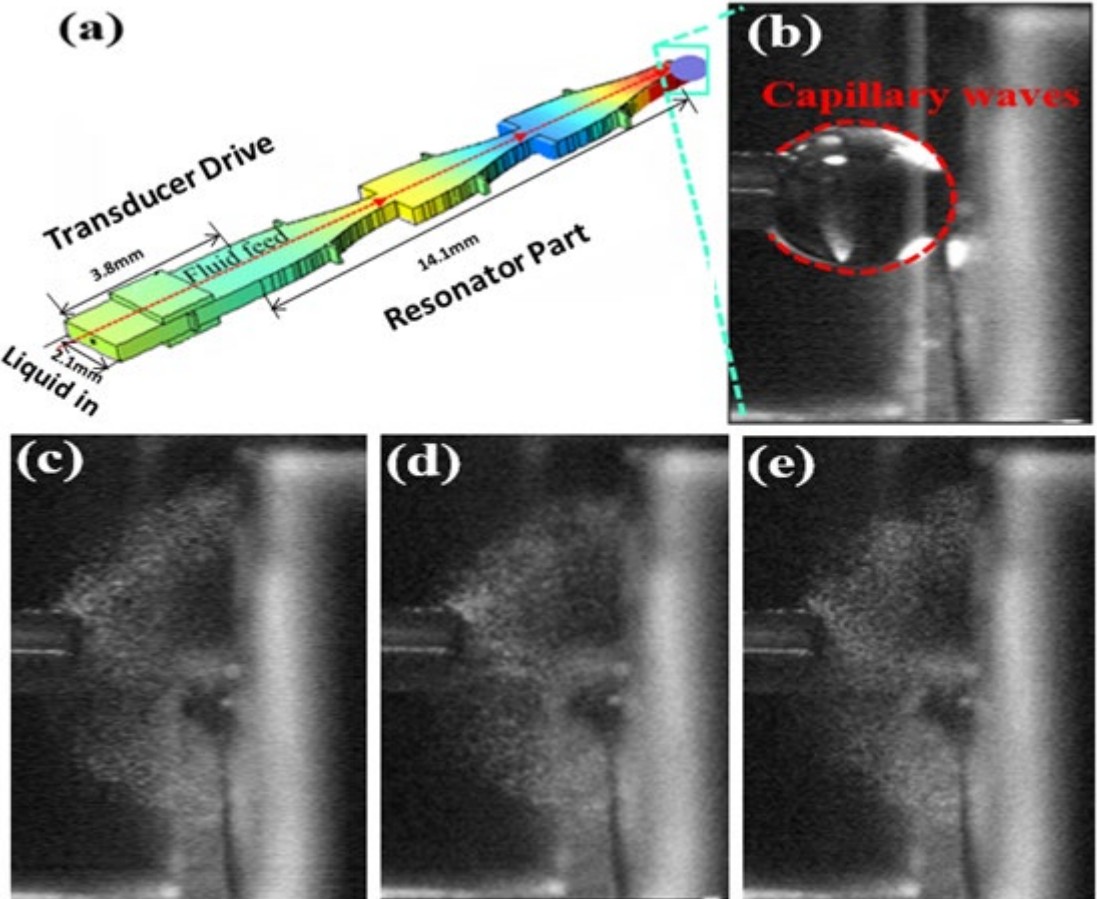

**Figure 5.** (**a**) Schematic presentation of a silicon-based three-horn ultrasonic nozzle. Optical images obtained from an atomization experiment after voltage was applied for (**b**) 6.49 s, (**c**) 6.50 s (**d**) 43.91 s, and (**e**) 63.59 s. The operating frequency was 484.5 kHz and the applied voltage was 7.9 V under 10 µL/min liquid flow.

Three gatherings of silicon wafers were examined for the sufficiency check of the longitudinal vibration frequency. Batches were named groups A, B, and C. Arabic numerals address the conveyed Fourier horn nozzle on a comparable silicon wafer. The wafer size was 4 inches in diameter and 530 µm in thickness. The speculative atomization (i.e., longitudinal vibration) was at 485 ± 1 kHz [26]. The cover design was made for the theoretical atomization reason. The sensible results are listed in Table 1. A schematic chart of the arrangement for estimating the longitudinal vibration at the spout tip appears in Figure 5. The pair of PZTs of the silicon-based, ultrasonic spout is driven by the exchanging flow (AC) electrical signal from an Agilent Function Generator Model 33120 A after enhancement by Amplifier Research Model 75 A250 (Amplifier Exploration, Souderton, PA). An Agilent 2-channel Oscilloscope 54621A screens the AC drive signal and gives an outside setting off of the Polytec Ultrasonics Vibrometer Controller Model OFV2700 (Polytech GmbH, Waldbronn, Germany). The regulator is essential for the Polytec laser Doppler vibrometer (LDV) that additionally contains a Polytec Fiber Interferometer Model OFV 511.

**Table 1.** Comparison of nozzle and atomization parameters. Vp shown is based on the best conditional atomization. Liquid flow rates were carefully adjusted to determine the potential operating conditions.

| Nozzle Number | A1 | A2 | A3 | B1 | B2 | B3 | C1 | C2 | C3 | C4 | C5 |
|---|---|---|---|---|---|---|---|---|---|---|---|
| Atomization Frequency (kHz) | 484.5 | 486.5 | 485 | 486 | 484.5 | 485 | 488 | 489 | 489 | 489 | 489 |
| Vp (V) | 7.9 | 6.5 | 8.7 | 7.9 | 6.9 | 6.5 | 7.1 | 6.5 | 6.6 | 6.5 | 6.8 |
| Liquid Flow Rate (µL/min) | 10–20 | 10–30 | 10–230 | 10–30 | 10–20 | 10–20 | 10–30 | 10–100 | 10–100 | 10–100 | 10–50 |

The test bar in a solitary fiber optic link is engaged and coordinated to the vibrating spout tip surface at a typical frequency. The photograph locator estimates the time subordinate force of the blended light of the test and the reference emissions. The subsequent (beat) recurrence of the blended light is only the Doppler recurrence shift that is relative to the tip vibration speed along with the pivot of the test shaft. The atomization frequency as the nozzle rotates is deliberately estimated by the Polytec LDV at driving voltage Vp before the atomized droplets are consistently and tenaciously encircled at the nozzle head. Note that each reverberating frequency of the instruments is recorded to the extent of 484.5–489 kHz. The differentiation in frequency is an immediate consequence of the holding deviation between any two silicon wafers set up together as resonators, as for a comparative wafer, and slight misalignment in the collecting methodology [31]. The liquid stream rate is also affected by collection error. A higher liquid stream rate (~100 µL/min) was tested and suffered due to MEMS nozzle fabrication.

A schematic diagram for measuring the longitudinal vibration at the tip of the nozzle is shown in Figure 6. The pair of PZTs of the silicon-based ultrasonic nozzle are driven by the AC electrical signal from an Agilent Function Generator 33120 A after amplification by Amplifier Research (AR) Model 75A250 (Amplifier Research, Souderton, PA). A 2-channel 54621A oscilloscope monitors the AC drive signal and provides an external triggering of the Polytec Vibrometer Controller Model OFV2700 (Polytech GmbH, Waldbronn, Germany) [26]. The controller is part of the Polytec laser Doppler vibrometer (LDV) that also contains a Polytec Fiber Interferometer Model OFV 511. A He-Ne laser at 632.8 nm wavelength in the interferometer is divided into a reference beam and a probe beam. The probe beam traveling in a single fiber optic cable is focused and directed to the vibrating nozzle tip surface at normal incidence. The photodetector measures the time-dependent intensity of the mixed light of the probe and the reference beams. The resulting (beat) frequency of the mixed light is just the Doppler frequency shift that is proportional to the tip vibration velocity along the axis of the probe beam [36]. Longitudinal vibration at the nozzle base also was measured to provide a reference for experimental determination of the vibration amplitude magnification (or gain) at the nozzle tip.

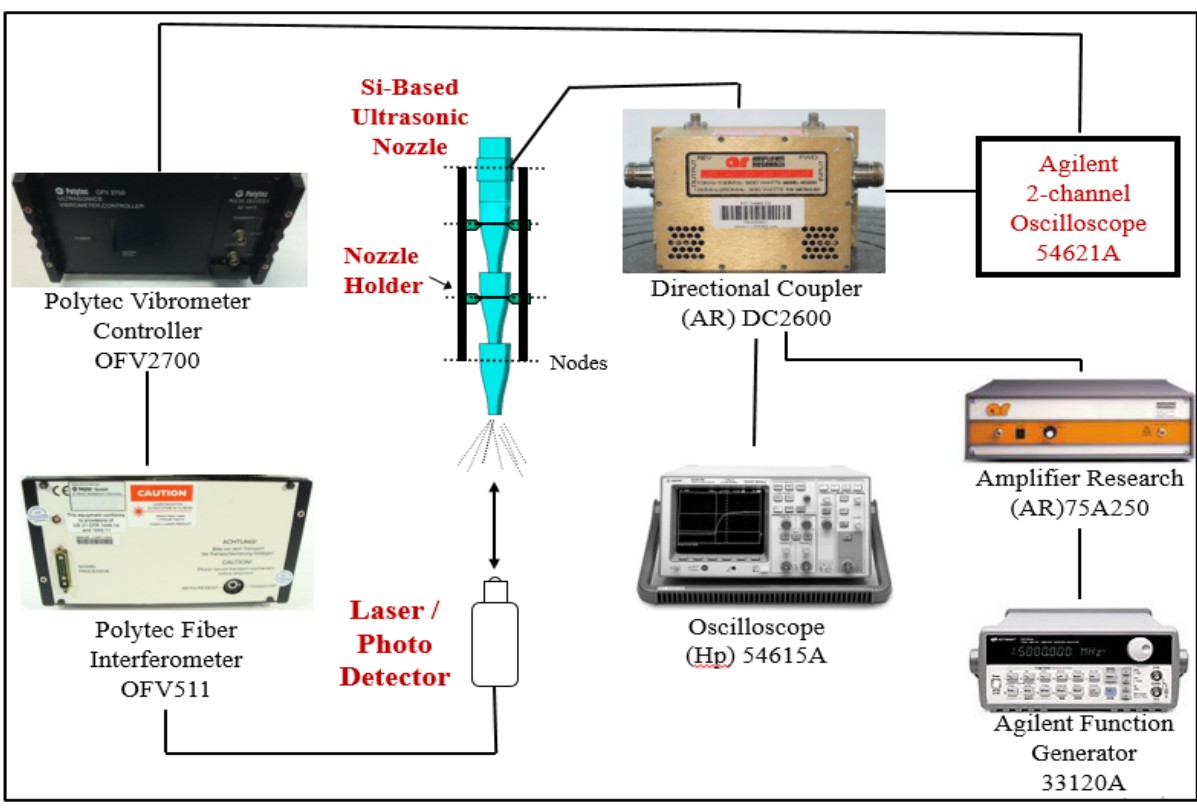

**Figure 6.** Schematic diagram for measuring the longitudinal vibration at the nozzle tip [26].

In Figure 7, the results are depicted in blue color. The calibration line is a function of the electrode voltage. The red coordinate (5.96, 9.125) on the calibration line is the suggested onset vibrational amplitude of 0.365 μm as the critical longitudinal displacement of capillary waves from modeling. The red coordinate (6.43, 10) confirms the linear regression of the applied electrode voltage for the Polytec LDV output signal in experiments. Note that the Polytec LDV output signal 1 V = 0.04 μm longitudinal displacement.

In Figure 7, the yield signal from the Polytec LDV estimations of the longitudinal vibrations of the three-horn ultrasonic nozzle at various driving voltages is shown. The straight backslide shows that the vibrational plenteousness of the response variable can be constrained by the working voltage. The developed change line predicts the longitudinal evacuating of the silicon-based three-horn Fourier nozzle as shown by the applied electrode voltage, similarly to the vibrational sufficiency. The exploratory result gives affirmations of the starting adequacy of capillary surface wave in high kHz for ultrasonic atomization. It can be noticed that the base applied electrode voltage of 5.96 V is adequate to produce strong capillary surface waves that can atomize deionized water at the nozzle tip. The details were elucidated in the modeling. The required minimum energy to atomize liquid drops is verified, and the modeling results correspond well to the experimental observations.

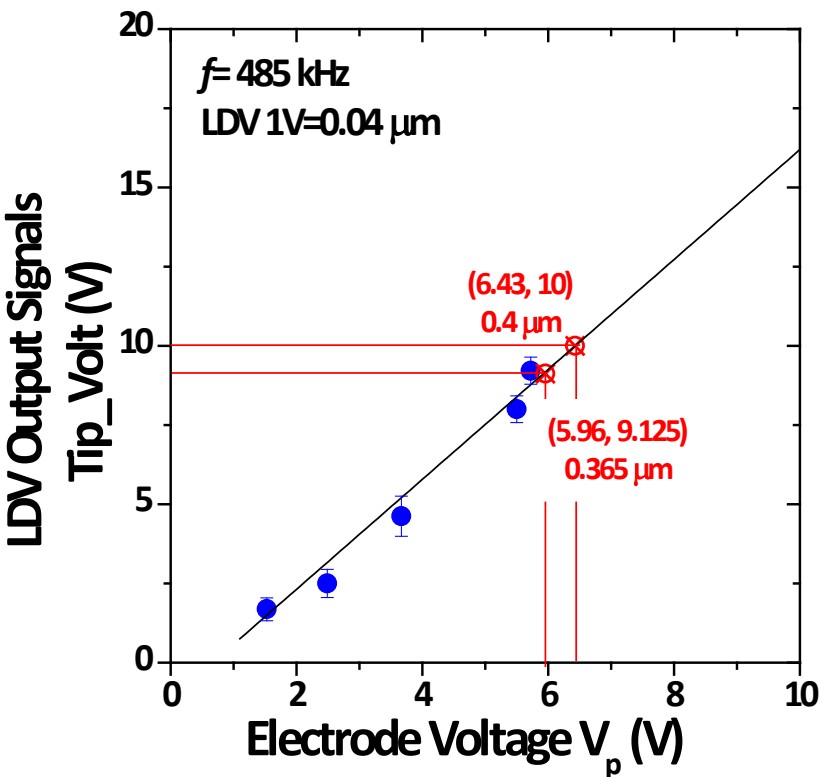

**Figure 7.** Output signals of longitudinal vibration at the nozzle tip versus applied electrode voltage at the base of the silicon-based three-horn Fourier nozzle.

## 5. Conclusions

This study of ultrasonic atomization of fluids for onset amplitude of initial capillary surface wave phenomena reports simulation results along with experimental verification. COMSOL Multiphysics 5.4 modeling predicts the threshold vibrational amplitude of 0.365 μm at 485 kHz. The constraint of the surface tension breaks when the amplitude of the capillary surface wave is increased as time elapses. Based on the basic physics that describes the water droplet formation process, the atomization of fluid through ultrasound waves in a piezoelectrically actuated nozzle was performed to determine the critical surface velocity of atomized droplets. Experimental studies prove that the surface plane with standing capillary surface waves would produce microdroplets at a designated electrode voltage of 5.96 V. The output voltage of longitudinal vibration at the nozzle bottom scales the driving voltage, and the frequency of microdroplets scales with capillary time, showing that silicon-based three Fourier-horn ultrasonic nozzles conform to the design concept from simulations. The advanced design of high-frequency ultrasonic nozzles demonstrated the minimum amplitude requirement of longitudinal vibration for the nozzle tip. The silicon-based three-horn Fourier nozzle is not only compact but also an energy-saving device. It promotes the further development of single-frequency ultrasonic nebulizers, especially in the green energy industry.

**Supplementary Materials:** The following are available online at www.mdpi.com/article/10.3390/w13141972/s1, Video S1: Velocity-0.40um.avi, www.mdpi.com/xxx/s2, Video S2: Amplitude-0.40um.avi, www.mdpi.com/xxx/s3, Video S3: 500kHz-Displacement.avi, www.mdpi.com/xxx/s4, Video S4: Velocity-0.36um.avi, www.mdpi.com/xxx/s5, Video S5: Amplitude-0.36um.avi, www.mdpi.com/xxx/s6, Video S6: B1200_3H3.avi.

**Author Contributions:** Conceptualization, Y.-L.S.; investigation, Y.-L.S. and C.-H.C.; methodology, Y.-L.S. and C.-H.C.; validation, C.-H.C.; writing—review and editing, C.-H.C. and Y.-L.S.; software, M.K.R.; data curation, M.K.R.; writing—original draft, M.K.R.; funding acquisition, Y.-L.S. All authors have read and agreed to the published version of the manuscript.

**Funding:** Ministry of Science and Technology, Taiwan, through grant MOST 106-2221-E-468-015 and MOST 110-2622-E-468-002.

**Institutional Review Board Statement:** Not applicable.

**Informed Consent Statement:** Not applicable.

**Data Availability Statement:** The study did not report any data.

**Acknowledgments:** This work was supported in part by the Ministry of Science and Technology, Taiwan, through grant MOST 106-2221-E-468-015.

**Conflicts of Interest:** The authors declare no conflict of interest.

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
