# Peer review of "Numerical Analysis of Ultrasonic Nebulizer for Onset Amplitude of Vibration with Atomization Experimental Results"

_water, doi:10.3390/w13141972_

Round 1

Reviewer 1 Report

The manuscript “Numerical Analysis of Ultrasonic Nebulizer for Onset Amplitude of Vibration with Atomization Experimental Results” deals with the design and characterization of an ultrasonic atomizer. The work reports numerical simulations of the atomizing system and subsequent experiments to demonstrate its effectiveness.

Atomizing systems are often employed in several engineering fields including drug delivery and spray atomizers, which makes the present research quite interesting. However, as pointed by the same authors at the very beginning of the paper, a huge number of investigations involved piezoelectric atomizers and/or capillary instability of a liquid jet. Therefore, the present reviewer’s recommendation is that it could be appropriate to discuss in more detail which advancements with respect to the previous literature are achieved within the present work.

As regards the computational part of the paper, the author should add more details about the simulations setup: which is the computational setup (are they solving just for the liquid inside the atomizer? Are they 2d or 3d computations? How do they impose the fluid motion? On which numerical schemes are relying to solve the equations written in the manuscript?) and also define the reference domain used in their paper (e.g. the “z axis” introduced in Fig. 1). Part of this information is already included into the supplementary material, but the present reviewer’s opinion is that they would better fit within the manuscript.

Minor comments are outlined below:

- line 95: (p,q) are never defined, perhaps it could be necessary to include Benjamin and Ursell’s stability chart within the present manuscript;

- line 106: nu is not the frequency, is the kinematic viscosity of the fluid

Reviewer 2 Report

The subject of paper is about the onset amplitude of the initial capillary surface wave for ultrasonic atomization of fluids. Although the contents of the manuscript are interesting, it needs to consider the following comments for publication.
1. Firstly, the literature review must be completed.  In Section 1, "the numerical and experimental research status should be clarified".
2. Are there any other non-dimension numbers related in this process?
3. Compared the calculation results with the experimental data and other reference simulations, the relative error and the reason that leads to the error in each graph should be better added.

Reviewer 3 Report

The paper (Numerical Analysis of Ultrasonic Nebulizer for Onset Amplitude of Vibration with Atomization Experimental Results) under review deals with the research on ultrasonic atomization. In the study, the onset amplitude of the initial capillary surface wave for ultrasonic atomization of fluids has been implemented. The results can be important from the point of view of the nebulization and drug delivery systems, The authors show models and results. The structure of the paper is in accordance with the principles of scientific reports. The article contains adequate and appropriately selected 36 literature items. In my opinion, the paper can be accepted for publication in the journal, although a medical journal would be a better choice.

Short comments:

  • Please, add an additional comments about practical aspects of the study (practical use of the results of the study).
  • Different sizes of fonts in the manuscript should be corrected.
  • Fig. 4 – the scale in mm should be added.
  • Is any statistical analysis of the experimental data (data presented in Fig. 6) eg. the standard deviation or standard error of estimation?

The above remarks do not diminish the importance of the manuscript but are intended to significantly increase its scientific value.
